# The Maternal Psychic Impact of Infection by SARS-CoV-2 during Pregnancy: Results from a Preliminary Prospective Study

**DOI:** 10.3390/healthcare12090927

**Published:** 2024-04-30

**Authors:** Lamyae Benzakour, Angèle Gayet-Ageron, Manuella Epiney

**Affiliations:** 1Department of Psychiatry, Geneva University Hospitals, 1205 Geneva, Switzerland; 2Faculty of Medicine, University of Geneva, 1206 Geneva, Switzerland; angele.gayet-ageron@unige.ch (A.G.-A.); manuela.epiney@unige.ch (M.E.); 3Division of Clinical Epidemiology, Department of Health and Community Medicine, Geneva University Hospitals, University of Geneva, 1206 Geneva, Switzerland; 4Department of the Woman, the Child and the Teenager, Geneva University Hospitals, 1205 Geneva, Switzerland

**Keywords:** pregnancy, SARS-CoV-2, mental health issues

## Abstract

Due to a higher risk of maternal complications during pregnancy, as well as pregnancy complications such as stillbirth, SARS-CoV-2 contamination during pregnancy is a putative stress factor that could increase the risk of perinatal maternal mental health issues. We included women older than 18 years, who delivered a living baby at the Geneva University Hospitals’ maternity wards after 29 weeks of amenorrhea (w.a.) and excluded women who did not read or speak fluent French. We compared women who declared having had COVID-19, confirmed by a positive PCR test for SARS-CoV-2, during pregnancy with women who did not, both at delivery and at one month postpartum. We collected clinical data by auto-questionnaires between time of childbirth and the third day postpartum regarding the occurrence of perinatal depression, peritraumatic dissociation, and peritraumatic distress during childbirth, measured, respectively, by the EPDS (depression is score > 11), PDI (peritraumatic distress is score > 15), and PDEQ (scales). At one month postpartum, we compared the proportion of women with a diagnosis of postpartum depression (PPD) and birth-related posttraumatic stress disorder (CB-PTSD), using PCL-5 for CB-PTSD and using diagnosis criteria according DSM-5 for both PPD and CB-PTSD, in the context of a semi-structured interview, conducted by a clinician psychologist. Off the 257 women included, who delivered at the University Hospitals of Geneva between 25 January 2021 and 10 March 2022, 41 (16.1%) declared they had a positive PCR test for SARS-CoV-2 during their pregnancy. Regarding mental outcomes, except birth-related PTSD, all scores provided higher mean values in the group of women who declared having been infected by SARS-CoV-2, at delivery and at one month postpartum, without reaching any statistical significance: respectively, 7.8 (±5.2, 8:4–10.5) versus 6.5 (±4.7, 6:3–9), *p* = 0.139 ***, for continuous EPDS scores; 10 (25.0) versus 45 (21.1), *p* = 0.586 *, for dichotomous EPDS scores (≥11); 118 (55.7) versus 26 (63.4), *p* = 0.359 *, for continuous PDI scores; 18.3 (±6.8, 16:14–21) versus 21.1 (±10.7, 17:15–22), 0.231 ***, for dichotomous PDI scores (≥15); 14.7 (±5.9, 13:10–16) versus 15.7 (±7.1, 14:10–18), *p* = 0.636 ***, for continuous PDEQ scores; 64 (30.0) versus 17 (41.5), *p* = 0.151 *, for dichotomous PDEQ scores (≥15); and 2 (8.0) versus 5 (3.6), *p* = 0.289 *, for postpartum depression diagnosis, according DSM-5. We performed Chi-squared or Fisher’s exact tests, depending on applicability for the comparison of categorical variables and Mann–Whitney nonparametric tests for continuous variables; *p* < 0.05 was considered as statistically significant. Surprisingly, we did not find more birth-related PTSD as noted by the PCL-5 score at one month postpartum in women who declared a positive PCR test for SARS-CoV-2:15 (10.6) versus no case of birth related PTSD in women who were infected during pregnancy (*p* = 0.131 *). Our study showed that mental outcomes were differently distributed between women who declared having been infected by SARS-CoV-2 compared to women who were not infected. However, our study was underpowered to explore all the factors associated with psychiatric issues during pregnancy, postpartum, depending on the exposure to SARS-CoV-2 infection during pregnancy. Future longitudinal studies on bigger samples and more diverse populations over a longer period are needed to explore the long-term psychic impact on women who had COVID-19 during pregnancy.

## 1. Introduction

Since the beginning of the COVID-19 pandemic, several studies have provided growing evidence of an increased risk of maternal mortality and morbidity as a higher risk for admission to the intensive care unit (ICU) for acute respiratory distress syndrome (ARDS), and more adverse pregnancy outcomes such as preeclampsia, postpartum hemorrhage, preterm birth, and stillbirth were described among pregnant women infected by SARS-CoV-2 [1,2,3].The mechanisms of these complications due to COVID-19 during pregnancy involve placental pathologies and immune responses at the maternal–fetal interface [2]. Nevertheless, outside the COVID-19 context, previous studies have suggested that pregnancy complications such as preeclampsia were risk factors for the occurrence of depression, as well as higher severe depressive symptoms [4], and more generally, that women classified as having medically moderate or high-risk pregnancy had higher incidence of anxiety disorders during pregnancy compared with women classified as experiencing a medically low-risk pregnancy [5]. Childbirth-related post-traumatic stress disorder (CB-PTSD) also appeared more frequently in women who experienced medical complications during pregnancy [6]. By analogy, we hypothesized that contamination by SARS-CoV-2 during pregnancy can induce a stress factor that could increase the risk of mental health issues in women during pregnancy and the postpartum period. To the best of our knowledge, there is no previous study assessing the impact of COVID-19 during pregnancy on the risk of developing CB-PTSD, postpartum depression, or other psychiatric issues postpartum. Indeed, several studies concluded the presence of a higher risk of mental issues during COVD-19. A study in China showed that pregnant patients assessed after the first wave of the COVID-19 pandemic had significantly higher rates of depression [7] (26.0% vs. 29.6%, *p* = 0.02) and higher rates of self-harm (*p* = 0.005) compared with women assessed before the start of the pandemic. This study highlighted that the rates of depression correlated positively with the number of confirmed cases of coronavirus disease (*p* = 0.003) and deaths per day (*p* = 0.001). The authors of this study concluded that low body mass index, primiparity, and age increased the risk of developing anxiety or depression during pregnancy in the context of the COVID-19 pandemic. A German study also noted an increased rate of stress and depression in women who delivered during the first wave of COVID-19 and who had been infected with SARS-CoV-2 during the postpartum period [8]. In a study conducted on primarily American women, the authors concluded that women giving birth during the pandemic had an increased rate of traumatic childbirth and birth-related PTSD compared with women giving birth before the pandemic [9]. None of the studies showing a negative impact of the COVID-19 pandemic on the maternal perinatal mental health explored the infection by SARS-CoV-2 during pregnancy.

The original aim of our research was to describe the prevalence of traumatic childbirth and CB-PTSD at one month postpartum in a prospective cohort of women who delivered at our hospital, and these data were previously published [10]. Because our initial study was conducted during the pandemic, we proposed an ad hoc objective to compare the mental health outcomes among women who delivered a live baby, according to their positive/negative status for SARS-CoV-2 [10].

## 2. Materials and Methods

### 2.1. Study Design and Population

We conducted a prospective cohort study of women who delivered at the University Hospitals of Geneva between 25 January 2021 and 10 March 2022 to assess traumatic childbirth and CB-PTSD prevalence and their associated risk factors during the COVID-19 pandemic.

During the inclusion period, the COVID-19 pandemic was still consequent in Geneva, according to Swiss epidemiological reports [11]. For a general population estimated at 198,979 people, the Swiss epidemiological report indicated 173 cases/day over 7 days per 100,000 people, increasing (+3%), with a 7.4% positivity rate, showing stable but with low reliability, with 111 active COVID-19 patients hospitalized, 227 post-COVID-19 patients hospitalized, and 22 patients hospitalized in HUG intensive care units during the week beginning of our inclusion. At the end of the study, the weekly number of cases increased significantly, with 3882 cases (+40% in one week), which was considered as a new wave. The weekly incidence of positive cases in Geneva was 767 cases per 100,000 people (+40% in one week) [11]. Because of this persistency in the number of cases, measures of protection, such as wearing masks and respecting social distancing to prevent contamination by SARS-CoV-2, were maintained in Geneva and in the specific COVID-19 hospitalization units during the entire period of inclusion of our study.

More specifically, the beginning of the vaccination campaign against SARS-CoV-2 began in May 2021, for Swiss for pregnant women with chronical disease, and in September 2021, for all the pregnant women. We did not collect the status of the women in regards to their vaccination against SARS-CoV-2. Regarding the inclusion period from the 25 January 2021 to the 10 March 2022, even though we do not have the exact estimation of the proportion of vaccinated pregnant women in our sample, it was probably still few during a largest part of the inclusion period. Moreover, the Delta variant of COVID-19 appeared during the inclusion period in Geneva, and it has been associated with more severe pneumopathy for pregnant women in comparison with that of the wild type, as well as the alpha and omicron variants [12].

### 2.2. Eligibility Criteria

The results of the initial study have already been published [13]. To be included, woman had to be older than 18 years, to deliver a living baby at the Geneva University Hospitals’ maternity wards after 29 weeks of amenorrhea (w.a.), and to consent to participate. Women who did not read or speak fluent French were excluded. We conducted a physical interview in their rooms during their maternity stays between the delivery day and the third day postpartum to inform them, to check the inclusion criteria and to determine their eligibility. 

### 2.3. Data Collection Procedures

In this study, we developed an ad hoc goal regarding the association between the self-declaration of SARS-CoV-2 contamination during pregnancy and different mental outcomes at delivery and at one month postpartum to assess maternal psychiatric outcomes. We divided women who self-declared having had COVID-19, confirmed by a positive PCR test for SARS-CoV-2 during pregnancy, from women who did not report either COVID-19 nor a positive test for SARS-CoV-2 during their pregnancy, based on laboratory test and their medical file, to minimize potential biases in participant recruitment. Women completed a questionnaire at three days following delivery and at one month postpartum. In the first three days following delivery, we collected socio-demographic variables (age, current profession, marital status, nationality) and psychiatric and traumatic event history. We collected the modalities of delivery and maternal and neonatal complication from the medical files to consider potential confounding factors for postpartum depression and CB-PTSD. We excluded missing data to ensure completeness of the collected information.

### 2.4. Outcome Measures and Other Variables

We assessed perinatal depression using the Edinburgh Perinatal Depression Scale (EPDS), a self-questionnaire which was validated to screen perinatal depression [6] and translated and adapted into French [14]. The EPDS score was dichotomized at 11, with a score of 11 or higher being previously associated with a medium to high probability for depression [6]. We assessed dissociative reactions during childbirth using the Peritraumatic Dissociative Experiences Questionnaire (PDEQ) [15,16,17], and we dichotomized the scores between scores less than 15 and scores more than 15, knowing that a score superior to 15 is associated to a higher risk of developing CB-PTSD. We assessed peritraumatic reactions regarding childbirth using the French version of the Peritraumatic Distress Inventory (PDI) [18,19], with a cut-off score value of more than 15 associated with a higher risk of developing a CB-PTSD [15], leading to a dichotomous variable (negative if <15; positive if ≥15). We used a self-questionnaire that we created for our study precisely for the women that experienced a COVID-19 infection during pregnancy, including the number and nature of symptoms of COVID-19, using a continuous variable checked by the women in a proposed list of symptoms (sore throat, having trouble breathing, chest pain, fever, sudden loss of smell and/or taste, headaches, general weakness, feeling unwell, muscle pain, cold, nausea). Women who were infected during pregnancy were asked about their subjective assessment of the impact of the COVID-19 infection using Likert scales regarding (1) their pregnancy experience, and (2) complications of pregnancy. All the answers for the Likert scales of this questionnaire were dichotomized between no (corresponding to “not at all” or “a little”) and yes (corresponding to “moderately”, “a lot”, “extremely”).

During the second assessment at one month postpartum, to evaluate the intensity of birth-related PTSD, the women completed the French version of the PTSD Checklist for DSM-5(PCL-5), a self-reporting questionnaire focusing on the 20 DSM-5 symptoms of PTSD, [20,21], the score being used as a continuous variable and a dichotomous variable, with a cut-off at 31 (negative if ≤31, positive if >31).

We also invited the participants to participate in a clinical interview to confirm the presence of CB-PTSD and to evaluate post-partum depression (PPD), according to the criteria for DSM-5 [22], which led to dichotomous variables (presence or absence of the disorder). We validated the criteria for the presence of traumatic childbirth for CB-PTSD, if the last childbirth corresponded to Criterion A, according to the DSM-5 for traumatic events [22].

### 2.5. Statistical Analysis

Descriptive statistics were reported as follows: mean ± standard deviation (SD), median (p50), and interquartile range (p25–p75), for continuous variables; frequencies and relative proportions, for categorical variables. We performed Chi-squared or Fisher’s exact tests, depending on applicability criteria (all expected frequencies >5 for applying the Chi-squared test; otherwise, Fisher’s exact test), to compare the categorical variables by SARS-CoV-2 infection status; due to small numbers and skewed distributions, we performed a Mann–Whitney nonparametric test to compare continuous variables between both groups. All *p*-values below 0.05 were considered statistically significant. Statistical analyses were performed using STATA IC 17.0.

## 3. Results

### 3.1. Descriptive Analysis

#### 3.1.1. Study Sample

Of the 257 women included, 41 (16.1%) declared they had a positive PCR test for SARS-CoV-2 during their pregnancy (Table 1). Women with positive tests had an average of 4.6 (±SD 2.4, p50:5:p25:3–p75.:6) symptoms of COVID-19 (Table 1). None of the women were admitted to ICU, and none developed the severe respiratory form of COVID-19. Among infected women, a small proportion (14.6%) professed a link between their COVID-19 and their pregnancy experience.

#### 3.1.2. Descriptive Data of the Sample and Comparison of Women by SARS-CoV-2 Status

We describe the integral sample using sociodemographic and clinical variables (Table 2).

We compared groups of women by SARS-CoV-2 status during pregnancy in regards to modalities of delivery, neonatal and maternal complications, and previous traumatic psychiatric events history and psychiatric assessment (Table 3). We did not find any significant differences between groups, except that women who declared previous SARS-CoV-2 infection had more maternal complications (such as episiotomy, perineal tears, post-partum hemorrhage, preeclampsia, neurological issues linked with anesthesia) than women who did not (29.6% vs. 7.9%, Table 3). Maternal complications like preeclampsia and postpartum hemorrhage were clinically meaningful because this kind of maternal complication was described in mothers that were infected by SARS-CoV-2, but some other maternal complications, like neurological issues secondary to anesthesia, perineal tears, or other issues were less clinically meaningful in this sense [1,2,3]. Neonatal complications were not statistically different between women who reported a COVID-19 infection during pregnancy when compared to uninfected women.

Regarding psychiatric assessment, we did not find any significant differences between groups for outcomes measured at delivery (EPDS, PDI, PDEQ) or for outcomes measured at one month postpartum (PCL-5, CB-PTSD, and postpartum depression, according to DSM-5) (Table 2).

## 4. Discussion

We proposed an ad hoc objective to our original study by comparing mental health outcomes during pregnancy and at one month postpartum according to the SARS-CoV-2 status during pregnancy. We found a higher rate of maternal complications in the literature, and in our study, we confirmed that the proportion of maternal complications was significantly higher among women infected compared to uninfected by SARS-CoV-2. However, the relatively small number of events made statistical adjustment for confounders inappropriate. We did not find any differences regarding the occurrence of psychiatric issues during pregnancy and at one month postpartum between infected and uninfected women.

There is a discrepancy between the medical perspective, which is aware of potential maternal and child complications, and the subjective maternal experience. Our findings are contradictory with the results of a German study that enrolled women who had given birth during the first wave of COVID-19 [8]. Indeed, unlike our study, this German study concluded that there was an association between concern regarding the COVID-19 pandemic and stress and depression, but none of the women included was infected by SARS-CoV-2, and the women were recruited during the first wave of COVID-19, while the women in our study were recruited well after the first wave of the COVID-19 pandemic [8]. We can explain these differences by the fact that pregnant women were more anxious and developed more psychiatric issues in the beginning of the COVID-19 pandemic than later in the course of the disease, but this hypothesis should be assessed in a comparative study between the different waves of COVID-19. We did not confirm an increased risk of antenatal depression in pregnant women during COVID-19, as some authors of a Chinese study did, in comparison with before COVID-19, but these authors did not explore the impact of a SARS-CoV-2 infection during pregnancy nor in connection with postpartum depression [7]. Our study had a different design and did not rigorously assess mental status during pregnancy to look for antenatal depression [7].We did not confirm the results from an American study that reported a higher rate of traumatic births and CB-PTSD in comparison with women that gave birth before the COVID-19 pandemic, but there was no information about SARS-CoV-2 contamination during pregnancy, and these results could have been linked only with the stress factors related to the COVID-19 pandemics [9]. Moreover, this study used an assessment by internet that may have introduced bias towards more educated women, and reliance on retrospective self-reported assessments may entail recall bias [9].

Our findings should be interpreted with caution owing to our small sample size and the low prevalence of outcomes, which may have reduced the statistical power to detect significant differences between groups. Second, we provided crude associations due to the small number of events that pertained to the performance of statistical adjustment for the known risk factors of traumatic childbirth related to childbirth PTSD. Third, we did not control for vaccination against SARS-CoV-2, which was initiated during the study, and for which we did not systematically collect data. Our results are in contrast with numerous data focusing on psychiatric issues following SARS-CoV-2 in contexts other than pregnancy that show that rates of depression (12%, 7–21%) and anxiety (23%, 13–33%) increased during this period [23]. Another limitation is that we did not include women who experienced stillbirth linked with COVID-19 in regards to the inclusion criteria for women delivering a living baby at the Geneva University Hospitals’ maternity wards after 29 weeks of amenorrhea (w.a.). Therefore, the psychiatric impact in the case of stillbirth secondary to termination or miscarriage linked to COVID-19 is probably different for these women.

Nevertheless, we mainly possess crude estimates of the link between COVID-19 infection and each outcome, due to the low number of events; our results must therefore be interpreted with great caution.

Finally, maternal psychiatric outcomes should have varied across different phases of the pandemic and as a result of the vaccination against SARS-CoV-2. The results of our study should have been different if we had assessed women during the first wave of the COVID-19 pandemic, and perhaps the maternal psychiatric outcome would have shown higher rates of CB-PTSD and DPP because of a higher perceived risks for maternal and neonatal complications in the beginning of COVID-19 pandemic.

Additional studies are needed to explore the potential impact of SARS-CoV-2 infection during pregnancy on psychiatric issues during postpartum, and these must include the potential impact of vaccination status regarding SARS-CoV-2 to confirm the preliminary findings in the present study.

## 5. Conclusions

There are few data regarding the impact of COVID-19 infection on maternal psychiatric issues. Our study showed higher proportions of CB-PTSD, antenatal depression, and postpartum depression among women infected by COVID-19 during pregnancy, but due to lack of strength of the study, the associations were not statistically significant. However, this study provides a preliminary descriptive picture that requires confirmation by larger studies considering different time periods of the pandemic.

## Figures and Tables

**Table 1 healthcare-12-00927-t001:** Answers related to COVID-19 during pregnancy.

Variables	
Positive test during pregnancy, N (%)	
No	213 (83.9)
Yes	41 (16.1)
Mean number of symptoms among infected patients (±SD, median:interquartile range)	4.6 (±2.4, 5:3–6)
Do you think that COVID infection had a negative impact on your pregnancy experience? N (%)	
Not at all/A little	35 (85.3)
Moderately/A lot/Extremely	6 (14.6)

**Table 2 healthcare-12-00927-t002:** Sociodemographic data, psychiatric and traumatic events history of all the women included (n = 254).

Variables	
Age (n = 247, 8 missing)	
Mean (min–max)	34.2 ± 4.8 (21–51)
Current profession (n = 230, 24 missing)	n (%)
Part-time job/Full-time job/in training	196 (85.2)
No job/disability status/on prolonged sick leave	34 (14.8)
Marital status (n = 244, 10 missing)	
Married or in a stable relationship	228 (93.4)
Single	16 (6.6)
Nationality (n = 254)	
Swiss	124 (48.8)
Other	130 (51.2)
Psychiatric history (n = 254)	
During your life, did you ever see a mental health professional?	
No	91 (41.4)
Yes	129 (58.6)
Were you ever hospitalized in psychiatry?	
No	214 (97.3)
Yes	−6 (2.7)
Were you ever treated by psychotropes?	
No	165 (75.0)
Yes	55 (25.0)
Previous traumatic events (n = 221, 33 missing)	
Exposure once or more than once to a traumatic event of any type	
No	162 (73.3)
Yes	59 (26.7)
If yes, nature of traumatic events?	
Physical aggression	16 (7.1)
Sexual aggression	29 (12.9)
Accident	13 (5.8)
Natural disaster	2 (0.9)
Attack	1 (0.4)
Severe disease	18 (8.0)
If yes, was it linked to a history of traumatic delivery?	
No	49 (83.0)
Yes	10 (17.0)
If yes, was it directly linked with several traumatic events?	
No	19 (32.2)
Yes	40 (67.8)
If yes, were you a direct witness to one or several traumatic events of others?	
No	37 (62.7)
Yes	22 (37.3)
If yes, was it linked to one or several traumatic events in your family members or beloved ones?	
No	40 (67.8)
Yes	19 (32.2)
If yes, were you exposed repeatedly or extremely frequently to traumatic events?	
No	48 (81.4)
Yes	11 (18.6)

**Table 3 healthcare-12-00927-t003:** Comparison between pregnant women who experienced COVID-19 during their pregnancy and those who did not. Prospective cohort study of women who delivered at the University Hospitals of Geneva between 25 January 2021 and 10 March 2022.

Variables	No COVID-19 during Pregnancy (n = 213)	COVID-19 during Pregnancy (n = 41)	*p*-Value
Modalities of delivery			
Instrumented vaginal delivery, n (%)	37 (24.0)	3 (11.1)	0.136 *
Spontaneous vaginal delivery, n (%)	120 (77.4)	18 (66.7)	0.252 **
Elective caesarean	18 (11.6)	3 (11.1)	
Emergency caesarean	17 (11.0)	6 (22.2)	
Neonatal complications ^1^, n (%)			0.593 **
No	147 (96.1)	27 (100.0)	
Yes	6 (3.9)	0 (0)	
Maternal complications ^2^, n (%)			0.004 **
No	140 (92.1)	19 (70.4)	
Yes	12 (7.9)	8 (29.6)	
Psychiatric assessment			
Perinatal depression using EPDS within three days post-partum, mean (±SD, p50:p25–p75)	6.5 (±4.7, 6:3–9)	7.8 (±5.2, 8:4–10.5)	0.139 ***
Perinatal depression using EPDS within three days post-partum, n (%)			0.586 *
<11	168 (78.9)	30 (75.0)	
≥11	45 (21.1)	10 (25.0)	
Peritraumatic distress at delivery using PDI, mean (±SD, p50:p25–p75)	18.3 (±6.8, 16:14–21)	21.1 (±10.7, 17:15–22)	0.231 ***
Peritraumatic distress at delivery using PDI, n (%)			0.359 *
<15	94 (44.3)	15 (36.6)	
≥15	118 (55.7)	26 (63.4)	
Peritraumatic dissociation at delivery using PDEQ, mean (±SD, p50:p25–p75)	14.7 (±5.9, 13:10–16)	15.7 (±7.1, 14:10–18)	0.636 ***
Peritraumatic dissociation at delivery using PDEQ, n (%)			0.151 *
<15	149 (70.0)	24 (58.5)	
≥15	64 (30.0)	17 (41.5)	
PCL-5 at one month, mean (±SD, p50:p25–p75, n)	10.3 (±10.7, 6:3–14, 179)	12.4 (±11.5, 8:5–19, 33)	0.247 ***
PCL-5 at one month, n (%)			0.999 *
<31	166 (92.7)	31 (93.9)	
≥31	13 (7.3)	2 (6.1)	
Birth-related PTSD according to DSM-5 at one month, n (%)			0.131 *
No	126 (89.4)	24 (100)	
Yes	15 (10.6)	0 (0)	
Post-partum depression according to DSM-5 at one month, n (%)			0.289 *
No	134 (96.4)	23 (92.0)	
Yes	5 (3.6)	2 (8.0)	

* Chi-squared test; ** Fisher’s exact test; *** Mann–Whitney nonparametric test. ^1^ Neonatal hypoxia, prematurity, fetal growth restriction. ^2^ Postpartum hemorrhage, preeclampsia, neurological issues secondary to anesthesia, perineal tears, or other issues.

## Data Availability

The data presented in this study are available on request from the corresponding author, due to restrictions for privacy and ethical considerations. The data are not publicly available due to the collected data being related to mailing addresses.

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
