# Peer review of "The Maternal Psychic Impact of Infection by SARS-CoV-2 during Pregnancy: Results from a Preliminary Prospective Study"

_healthcare, 2024, doi:10.3390/healthcare12090927_

Round 1

Reviewer 1 Report

Comments and Suggestions for Authors

1- in lines 75-78 you said:"We divided women who self-declared having had a positive PCR test for SARS-CoV-2 during pregnancy from women who did not report such event." Would you exclude from the control group a woman who had symptoms and did not perform PCR due to concern or any other reason?

2- What were your include and exclude criteria? Did you consider psychological factors related to pregnancy and postpartum blue, the severity of which is different in each person, as confounding?

3- According to the title of the study "The maternal psychic impact of infection by SARS-CoV-2 during pregnancy: results from a prospective study", it was expected that women who were infected with SARS-CoV-2 that have maternal or fetal complications or death, considered In the present study but in this study women with positive PCR did not mention complications in pregnancy, and there is a possibility that their psychological impact is different from those with complications.

Author Response

  • in lines 75-78 you said:"We divided women who self-declared having had a positive PCR test for SARS-CoV-2 during pregnancy from women who did not report such event." Would you exclude from the control group a woman who had symptoms and did not perform PCR due to concern or any other reason?

We thank you for your pertinent comment. The COVID-19 group women were women who had COVID-19 confirmed by a positive PCR test for SARS-CoV-2. In the control group, women did not declare COVID-19 and did not experienced a positive PCR test for SARS-CoV-2 during their pregnancy. We clarified this point in the revised manuscript (l.84): “We divided women who self-declared having had COVID-19 confirmed by a positive PCR test for SARS-CoV-2 during pregnancy from women who did not report neither COVID-19 nor a positive test for SARS-CoV-2 during their pregnancy.”

2- What were your include and exclude criteria? Did you consider psychological factors related to pregnancy and postpartum blue, the severity of which is different in each person, as confounding?

We thank you for your comment. We should have indeed described the inclusion and exclusion criteria that have been used for the original study, knowing that this present study is an ad-hoc one. We clarified the inclusion and exclusion criteria in the revised manuscript (cf. l.75-80): “To be included, woman had to be older than 18 years, to deliver a living baby at the Geneva University Hospitals’ maternity wards after 29 weeks of amenorrhea (w.a.) and to consent to participate. Women who did not read or speak fluent French were excluded. We conducted a physical interview in their room during maternity between the delivery day and the third day postpartum to inform them of and to check the inclusion criteria.”

We indeed consider psychological factors related to postpartum depression and CB-PTSD using different questionnaires assessing modalities of delivery, maternal and neonatal complications, psychic maternal factors. The assessment of these risk factors are described in the original manuscript and we clarified that they were confounding factors: “Women completed a questionnaire at three days following delivery and at one-month postpartum to consider potential confounding factors for postpartum depression and CB-PTSD Women completed a questionnaire at three days following delivery and at one-month postpartum to consider potential confounding factors for postpartum depression and CB-PTSD. In the first three days following delivery, we collected socio-demographic variables (age, current profession, marital status, nationality) and psychiatric and traumatic event history. We collected modalities of delivery, maternal and neonatal complication from medical file. We assessed perinatal depression using Edinburgh Perinatal Depression Scale (EPDS), a self-questionnaire which was validated to screen perinatal depression [6] and translated and adapted to French [7]. The EPDS was dichotomized at 11: having 11 or more has been previously associated with middle to high probability for depression [6]. We assessed dissociative reactions during childbirth using the Peritraumatic Dissociative Experiences Questionnaire (PDEQ) [8][9] [10] and we dichotomized the scores between scores less than 15 and scores more than 15, knowing that a score superior to 15 is associated to a higher risk of developing CB-PTSD. We assessed peritraumatic reactions around childbirth using the French version of the Peritraumautic distress Inventory (PDI) [11] [12] with a cut-off score value of more than 15 associated to a higher risk of developing a CB-PTSD [11], leading to a dichotomous variable (negative if <15 positive if ≥ 15). We used self-questionnaire that we created for our study to precise for the women that had a COVID-19 infection during pregnancy the number and nature of symptoms of COVID-19 using a continuous variable, the negative association or not of a COVID-19 infection on their pregnancy experience rated on Likert scales. During second assessment at one month postpartum, we assessed CB-PTSD by asking the women to complete the French version of the PTSD Checklist for DSM-5 (PCL-5) that is a self-report questionnaire included items of the PTSD Checklist for DSM-5, assessing the 20 DSM-5 symptoms of PTSD, which we used to evaluate the intensity of birth-related PTSD [13][14], the score being used as a continuous variable and a dichotomous variable with a cut-off at 31 (negative if ≤31,positive if >31). “

We did not consider postpartum blues because that is not considered as a confounding factor or a risk factor for postpartum depression nor for CB-PTSD. Postpartum blues is a non-psychiatric and physiological condition correlated to milky rise and provides transient and spontaneously resolutive psychic symptoms and cannot keep more than 7 days. 

Finally, we mentioned as a limitation in the discussion that we could not provide adjusted estimates due to the low number of events and small sample size in our study: “Nevertheless, we mainly have crude estimates between Covid-19 infection and each outcome, due to the low number of events; our results must therefore be interpreted with great caution.”

3-According to the title of the study "The maternal psychic impact of infection by SARS-CoV-2 during pregnancy: results from a prospective study", it was expected that women who were infected with SARS-CoV-2 that have maternal or fetal complications or death, considered In the present study but in this study women with positive PCR did not mention complications in pregnancy, and there is a possibility that their psychological impact is different from those with complications.

We thank you for your comment. Women who had COVID-19 during pregnancy declared maternal and neonatal complications. Complications during pregnancy could not be fetal death regarding the inclusion criteria for women to deliver a living baby at the Geneva University Hospitals’ maternity wards after 29 weeks of amenorrhea (w.a.). We added in the discussion this limitation:

“We did not include women who had stillbirth in link with COVID-19 regarding inclusion criteria for women to deliver a living baby at the Geneva University Hospitals’ maternity wards after 29 weeks of amenorrhea (w.a.). Therefore the psychiatric impact in the case of stillbirth secondary to termination or miscarriage in link with COVID-19 is probably different for these women.”

Reviewer 2 Report

Comments and Suggestions for Authors

The topic is interesting, however I do have a few remarks.

Even though the English is fine, there are a few minor mistakes such as in the materials and methods the spaces is missing "We assessed perinataldepression". And in the discussion "We found a higher rate of maternal complications in literaturer". I also suggest writing the abstract in a continuous form. Also, maybe mention the informed consent that the examinees had to sign.

Author Response

1-Even though the English is fine, there are a few minor mistakes such as in the materials and methods the spaces is missing "We assessed perinataldepression". And in the discussion "We found a higher rate of maternal complications in literaturer".

We thank you for your comments. We corrected these mistakes in the revised manuscript. (l.92, 166)

2- I also suggest writing the abstract in a continuous form.

We thank you for your suggestion. We deleted subtitles and wrote the abstract in a continuous form.

Abstract: Due to a higher risk of maternal complications during pregnancy, and pregnancy complications such as stillbirth, SARS-CoV-2 contamination during pregnancy is a putative stress factor that could increase the risk of perinatal maternal mental health issues.  We compared women who declared having had a positive PCR test for SARS-CoV-2 during pregnancy with women who did not at delivery and at one-month regarding the occurrence of perinatal depression between childbirth and the third day postpartum, peritraumatic dissociation and peritraumatic distress during childbirth (measured by the EPDS, PDI and PDEQ scales).At one-month, we compared the proportion of women who are with a diagnosis of postpartum depression (PPD) and birth-related posttraumatic stress disorder (PTSD) by PCL-5 and using diagnosis criteria according DSM-5. On the 257 women included who delivered at the University Hospitals of Geneva between 25 January 2021 and 10 March 2022, 41 (16.1%) declared they had a positive PCR test for SARS-CoV-2 during their pregnancy. Regarding mental outcomes except birth-related PTSD, all scores provided higher mean values in the group of women who declared having been infected by SARS-CoV-2 at delivery and at one-month without reaching any statistical significance: respectively (7.8 (±5.2, 8: 4-10.5) versus 6.5 (±4.7, 6: 3-9), p=.139***  for continuous EPDS scores,10 (25.0) versus 45 (21.1), p= 0.586*, for dichotomous EPDS scores(≥11), 118 (55.7) versus 26 (63.4), p=0.359*  for continuous PDI scores 18.3 (±6.8, 16:14-21) versus 21.1 (±10.7, 17:15-22), 0.231***,118 (55.7)versus 26 (63.4), p=0.359* for dichotomous PDI scores (≥15), 14.7 (±5.9, 13: 10-16) versus 15.7 (±7.1, 14: 10-18), p=0.636***, for continuous PDEQ scores, and 64 (30.0) versus 17 (41.5) , p=0.151*, for dichotomous PDEQ scores (≥15) and 2 (8.0)versus 5 (3.6), p= .289* for postpartum depression diagnosis according DSM-5. Surprisingly, we did not find more birth-related PTSD regarding PCL-5 score at 1 month in women who declared positive PCR test for SARS-CoV-2: 15 (10.6) versus no case of birth related PTSD in women who were infected during pregnancy (p=.131*). Our study showed that mental outcomes were differently distributed between women who declared having been infected by SARS-CoV-2 compared to women who were not infected. However, our study was underpowered to explore all the factors associated with psychiatric issues during pregnancy, postpartum depending on the exposure to SARS-CoV-2 infection during pregnancy.

3-Also, maybe mention the informed consent that the examinees had to sign.

We thank you for your comment. We had already a sentence about the informed consent l. 222

“Informed Consent Statement: Informed consent was obtained from all subjects involved in the study.”

Reviewer 3 Report

Comments and Suggestions for Authors

In the title should be added: results of a preliminary prospective study since the study sample is small

The study has significant limitations which interpret the conclusions as doubtful.

The one year examination is insufficient because mental outcomes after SARS-CoV-2 contamination can appear later.

Finally, the lack of vaccination status in the small positive sample could change the results of this study.      

Author Response

  1. In the title should be added: results of a preliminary prospective study since the study sample is small

We thank you for your comment. The sample is indeed small. Knowing that we did not plan a most important study on a larger sample, we did not consider to qualify the study as “preliminary”. Nevertheless, we can change the title as you suggest.

  1. The study has significant limitations which interpret the conclusions as doubtful.

Thank you for your comment that we share. Our significant limitations explain why we have to interpret carefully our results, We acknowledged it in the discussion and the conclusion in the original manuscript (l.185-203):

Our findings should be interpreted with caution owing to our small sample size and the low prevalence of outcomes, which may have reduced statistical power to detect significant differences between groups. Second, we provided crude associations due to the small number of events that have pertained the performance of statistical adjustment for known risk factors of traumatic childbirth related to childbirth PTSD. Third, we did not control for vaccination against SARS-CoV-2 which was organised during the study and for which we did not systematically collect data.  Our results are in contrast with the numerous data focusing on psychiatric issues following SARS-CoV-2 in other contexts than pregnancy that show that rates of depression (12%, 7%-21%) and anxiety (23%, 13%-33%) increased [17]. Another limitation is that we did not include women who had stillbirth in link with COVID-19 regarding inclusion criteria for women to deliver a living baby at the Geneva University Hospitals’ maternity wards after 29 weeks of amenorrhea (w.a.). The psychiatric impact in the case of stillbirth secondary to termination or miscarriage in link with COVID-19 is probably different for these women.

Nevertheless, we mainly have crude estimates between Covid-19 infection and each outcome, due to the low number of events; our results must therefore be interpreted with great caution.

 Further studies are needed to explore the potential impact of SARS-Cov-2 infection during pregnancy on psychiatric issues during post-partum and will have to include to potential impact of vaccination status regarding SARS-CoV-2.

  1. The one year examination is insufficient because mental outcomes after SARS-CoV-2 contamination can appear later.

Thank you for your comment. We think that you wanted to write "one month" that is the follow-up time in the present study instead of "one year"Our study aimed to .

assess the psychic impact at one month postpartum in women that presented COVID-19 at any time during pregnancy. The delay between the SARS-CoV-2 infection and the assessment at one month postpartum considerably varies among women that were infected because the infection could have been at any time during pregnancy. We choose to focus on the one-month postpartum mental state whatever the date of the infection because most of CB-PTSD and postpartum depression occur at this timepoint independently of COVID-19 and that is the usual timepoint of the studies for these disorders. Our hypothesis was that COVID-19 could play as a risk factor for CB-PTSD and postpartum depression. We did not look for an increased risk of post-COVID-19 condition that would have needed indeed specific delays with more than 3 months between the infection and the assessment according to the ICD definition (https://www.who.int/publications/i/item/WHO-2019-nCoV-Post_COVID-19_condition-Clinical_case_definition-2021.1 ): “Post COVID-19 condition occurs in individuals with a history of probable or confirmed SARS CoV-2 infection, usually 3 months from the onset of COVID-19 with symptoms and that last for at least 2 months and cannot be explained by an alternative diagnosis. Common symptoms include fatigue, shortness of breath, cognitive dysfunction but also others and generally have an impact on everyday functioning. Symptoms may be new onset following initial recovery from an acute COVID-19 episode or persist from the initial illness. Symptoms may also vary or relapse over time."

  1. Finally, the lack of vaccination status in the small positive sample could change the results of this study.      

Thank you for your comment that we absolutely share. We highlight the important limitation in line with the lack of vaccination status in our discussion in our original manuscript:

l.189-191 “Third, we did not control for vaccination against SARS-CoV-2 which was organised during the study and for which we did not systematically collect data.”

l.201-203 Further studies are needed to explore the potential impact of SARS-Cov-2 infection during pregnancy on psychiatric issues during post-partum and will have to include to potential impact of vaccination status regarding SARS-CoV-2.

Reviewer 4 Report

Comments and Suggestions for Authors

I have  comprehensive review of your paper. I have provided constructive feedback aimed at enhancing the quality and impact of your article. I encourage you to consider these suggestions to improve clarity and overall effectiveness. Upon implementing revisions, I am eager to review the updated version. Your commitment to refining your work is admirable, and I am eager to witness the evolution of your article. Best regards.

Abstract

    • Clearly outline the methodology employed in the study, including participant selection criteria, data collection methods, and measurement tools used for assessing mental health outcomes. The abstract mentions the EPDS, PDI, PDEQ scales, and DSM-5 criteria, but it lacks clarity on how these were applied.
    • Provide more context for the statistical analyses conducted and the interpretation of results. For example, explain why certain differences were considered statistically significant while others were not.
    • Offer suggestions for future research directions, including larger-scale studies with more diverse populations and longitudinal follow-up to assess longer-term mental health outcomes.

Introduction

    • Provide a clearer statement of the research gap or question that the study aims to address. What specific mental health issues associated with SARS-CoV-2 infection during pregnancy are being investigated?
    • Consider providing a brief overview of the existing literature on the mental health outcomes of pregnant women during the COVID-19 pandemic, including any gaps or limitations in current knowledge.
    • Provide more detailed background information on the observed risks and complications of SARS-CoV-2 infection during pregnancy. For example, you could elaborate on the mechanisms through which viral infection may impact maternal and fetal health.
    • Avoid introducing new concepts or objectives towards the end of the introduction. Instead, integrate them seamlessly within the main narrative.

method

    • . However, it could benefit from clearer organization and structure to enhance readability.
    • Consider breaking down the methods into subsections based on different aspects of the study protocol (e.g., participant recruitment, data collection procedures, outcome measures, statistical analysis).
    • Specify how potential biases in participant selection were addressed, especially concerning the inclusion of women with a positive PCR test for SARS-CoV-2 during pregnancy.
    • Describe the methods used to minimize missing data and ensure completeness of the collected information.
    • Clarify how SARS-CoV-2 infection during pregnancy was determined, particularly the criteria used for self-reporting and confirmation of positive PCR tests.
    • Describe the specific COVID-19 symptoms assessed in the questionnaire and how their severity or impact on pregnancy was evaluated.

results

    • The comparison between women with and without SARS-CoV-2 infection during pregnancy is presented systematically, which is commendable. However, ensure that the text provides a clear interpretation of the statistical findings, particularly regarding the significance levels and directionality of the associations.
    • Emphasize clinically meaningful differences between the groups, especially regarding maternal and neonatal complications and psychiatric outcomes.
    • Provide additional context or explanations for the statistical tests used to compare categorical and continuous variables between groups (e.g., Chi-square test, Mann-Whitney test).
    • Clearly specify the significance levels used for determining statistical significance (e.g., p < 0.05) and any adjustments made for multiple comparisons.

discussion

    • Clearly interpret the study findings in relation to the existing literature. Discuss how the observed rates of maternal complications and psychiatric issues among women infected with SARS-CoV-2 during pregnancy compare to previous studies and what implications these findings may have for clinical practice.
    • Address any discrepancies or contradictions between the current study and previous research, such as the differences in findings compared to the German study mentioned.
    • Acknowledge the limitations of the study, such as the small sample size and low prevalence of outcomes, which may have limited the statistical power to detect significant differences between groups.
    • Provide a comprehensive comparison with existing literature on the impact of COVID-19 infection on maternal psychiatric issues. Discuss how the findings of the current study align with or diverge from previous research in this area.
    • Highlight the importance of investigating psychiatric outcomes across different phases of the pandemic to understand how the evolving COVID-19 situation may influence maternal mental health.
    • Emphasize the need for caution in interpreting the results and the importance of further research to confirm and expand upon the preliminary findings presented in the current study.

Comments on the Quality of English Language

I have  comprehensive review of your paper. I have provided constructive feedback aimed at enhancing the quality and impact of your article. I encourage you to consider these suggestions to improve clarity and overall effectiveness. Upon implementing revisions, I am eager to review the updated version. Your commitment to refining your work is admirable, and I am eager to witness the evolution of your article. Best regards.

Abstract

    • Clearly outline the methodology employed in the study, including participant selection criteria, data collection methods, and measurement tools used for assessing mental health outcomes. The abstract mentions the EPDS, PDI, PDEQ scales, and DSM-5 criteria, but it lacks clarity on how these were applied.
    • Provide more context for the statistical analyses conducted and the interpretation of results. For example, explain why certain differences were considered statistically significant while others were not.
    • Offer suggestions for future research directions, including larger-scale studies with more diverse populations and longitudinal follow-up to assess longer-term mental health outcomes.

Introduction

    • Provide a clearer statement of the research gap or question that the study aims to address. What specific mental health issues associated with SARS-CoV-2 infection during pregnancy are being investigated?
    • Consider providing a brief overview of the existing literature on the mental health outcomes of pregnant women during the COVID-19 pandemic, including any gaps or limitations in current knowledge.
    • Provide more detailed background information on the observed risks and complications of SARS-CoV-2 infection during pregnancy. For example, you could elaborate on the mechanisms through which viral infection may impact maternal and fetal health.
    • Avoid introducing new concepts or objectives towards the end of the introduction. Instead, integrate them seamlessly within the main narrative.

method

    • . However, it could benefit from clearer organization and structure to enhance readability.
    • Consider breaking down the methods into subsections based on different aspects of the study protocol (e.g., participant recruitment, data collection procedures, outcome measures, statistical analysis).
    • Specify how potential biases in participant selection were addressed, especially concerning the inclusion of women with a positive PCR test for SARS-CoV-2 during pregnancy.
    • Describe the methods used to minimize missing data and ensure completeness of the collected information.
    • Clarify how SARS-CoV-2 infection during pregnancy was determined, particularly the criteria used for self-reporting and confirmation of positive PCR tests.
    • Describe the specific COVID-19 symptoms assessed in the questionnaire and how their severity or impact on pregnancy was evaluated.

results

    • The comparison between women with and without SARS-CoV-2 infection during pregnancy is presented systematically, which is commendable. However, ensure that the text provides a clear interpretation of the statistical findings, particularly regarding the significance levels and directionality of the associations.
    • Emphasize clinically meaningful differences between the groups, especially regarding maternal and neonatal complications and psychiatric outcomes.
    • Provide additional context or explanations for the statistical tests used to compare categorical and continuous variables between groups (e.g., Chi-square test, Mann-Whitney test).
    • Clearly specify the significance levels used for determining statistical significance (e.g., p < 0.05) and any adjustments made for multiple comparisons.

discussion

    • Clearly interpret the study findings in relation to the existing literature. Discuss how the observed rates of maternal complications and psychiatric issues among women infected with SARS-CoV-2 during pregnancy compare to previous studies and what implications these findings may have for clinical practice.
    • Address any discrepancies or contradictions between the current study and previous research, such as the differences in findings compared to the German study mentioned.
    • Acknowledge the limitations of the study, such as the small sample size and low prevalence of outcomes, which may have limited the statistical power to detect significant differences between groups.
    • Provide a comprehensive comparison with existing literature on the impact of COVID-19 infection on maternal psychiatric issues. Discuss how the findings of the current study align with or diverge from previous research in this area.
    • Highlight the importance of investigating psychiatric outcomes across different phases of the pandemic to understand how the evolving COVID-19 situation may influence maternal mental health.
    • Emphasize the need for caution in interpreting the results and the importance of further research to confirm and expand upon the preliminary findings presented in the current study.

Author Response

Dear reviewer,

We thank you for your detailed comprehensive review of your paper that will help us to improve our article. Here are our answers point by point.

Abstract

    • Clearly outline the methodology employed in the study, including participant selection criteria, data collection methods, and measurement tools used for assessing mental health outcomes. The abstract mentions the EPDS, PDI, PDEQ scales, and DSM-5 criteria, but it lacks clarity on how these were applied.

 We thank you for your comment. We clarified the methodology in the revised manuscript.

    • Provide more context for the statistical analyses conducted and the interpretation of results. For example, explain why certain differences were considered statistically significant while others were not.

We thank you for this suggestion. We made several changes in the revised manusctript .

    • Offer suggestions for future research directions, including larger-scale studies with more diverse populations and longitudinal follow-up to assess longer-term mental health outcomes.

We thank you for your comment. We completed the future research directions in the revised manuscript (l.48-50):

“Future longitudinal studies on bigger samples and more diverse populations on longer period are needed to explore long term psychic impact on women who had COVID-19 during their pregnancy.”

Introduction

    • Provide a clearer statement of the research gap or question that the study aims to address. What specific mental health issues associated with SARS-CoV-2 infection during pregnancy are being investigated?

We thank you for your comment. W tried to clarify in the introduction l.72-74

“To the best of our knowledge, there is no previous study assessing the impact of COVID-19 during pregnancy on the risk of developing CB-PTSD, postpartum depression or other psychiatric issues during postpartum. Indeed, several studies concluded to a higher risk of mental issues during COVD-19 but there is a lack of data exploring the existence of mental health issues specifically associated with the infection by SARS-CoV-2 during pregnancy.”

    • Consider providing a brief overview of the existing literature on the mental health outcomes of pregnant women during the COVID-19 pandemic, including any gaps or limitations in current knowledge.

We thank you for your comment.

    • Provide more detailed background information on the observed risks and complications of SARS-CoV-2 infection during pregnancy. For example, you could elaborate on the mechanisms through which viral infection may impact maternal and fetal health.
    • Thank you for your comment. We added a sentence about mechanisms underlying complications during pregnancy in link with COVID-19
    • Avoid introducing new concepts or objectives towards the end of the introduction. Instead, integrate them seamlessly within the main narrative.

method

    • . However, it could benefit from clearer organization and structure to enhance readability.

Thank you for your comment. We clarified the organization and structure in the revised manuscript.

    • Consider breaking down the methods into subsections based on different aspects of the study protocol (e.g., participant recruitment, data collection procedures, outcome measures, statistical analysis).

Thank you for your comment. We organized subsections.

    • Specify how potential biases in participant selection were addressed, especially concerning the inclusion of women with a positive PCR test for SARS-CoV-2 during pregnancy.

Thank you for comment. We added a sentence to clarify this point in the revised manuscript.

See l.133-136

“We divided women who self-declared having had COVID-19 confirmed by a positive PCR test for SARS-CoV-2 during pregnancy from women who did not report neither COVID-19 nor a positive test for SARS-CoV-2 during their pregnancy based on laboratory test and medical file to minimize potential biases in participant recruitment.”

    • Describe the methods used to minimize missing data and ensure completeness of the collected information.

We thank you for your comment. We excluded missing data to ensure completeness of the collected information. We added a sentence (See l.142-143)

    • Clarify how SARS-CoV-2 infection during pregnancy was determined, particularly the criteria used for self-reporting and confirmation of positive PCR testsi.

Thank you for your comment. We clarified this point in the revised manuscript (See l.133-137)

“We divided women who self-declared having had COVID-19 confirmed by a positive PCR test for SARS-CoV-2 during pregnancy from women who did not report neither COVID-19 nor a positive test for SARS-CoV-2 during their pregnancy based on laboratory test and medical file to minimize potential biases in participant recruitment.”

    • Describe the specific COVID-19 symptoms assessed in the questionnaire and how their severity or impact on pregnancy was evaluated.

 COVID-19 symptoms that we assessed were the more frequent symptoms described in COVID-19:□sore throat

□ Cough

□ Having trouble breathing

□ Chest pain

□ Fever

□ Sudden loss of smell and/or taste

□ Headaches

□ General weakness, feeling unwell

□ Muscle pain

□ Cold

□ Nausea

We added a sentence on this point in the revised manuscript.

We collected clinical informations regarding maternal, obstetrical and neonatal complications.

results

    • The comparison between women with and without SARS-CoV-2 infection during pregnancy is presented systematically, which is commendable. However, ensure that the text provides a clear interpretation of the statistical findings, particularly regarding the significance levels and directionality of the associations.

We verified all interpretation in the text and we noticed that we did not provide the correct Table number. All comparisons between women who declared having been infected vs. not are presented in Table 3 and not 2. Sorry for the confusion.

    • Emphasize clinically meaningful differences between the groups, especially regarding maternal and neonatal complications and psychiatric outcomes.

We thank you for your comment. There were indeed statistically significant differences between the groups regarding maternal complications (See Table 3), COVID-19 group presenting more maternal complications with a p=0.04. Maternal complications like preeclampsia and postpartum hemorrhage were clinically meaningful because this kind of maternal complications were described in mothers that were infected by SARS-CoV-2 but some other maternal complications like neurological issues secondary to anesthesia or perineal tears or other issues were less clinically meaningful in this sense. (Kumar D, Verma S, Mysorekar IU. COVID-19 and pregnancy: clinical outcomes; mechanisms, and vaccine efficacy. Transl Res. 2023 Jan;251:84-95. doi: 10.1016/j.trsl.2022.08.007. Epub 2022 Aug 12. PMID: 35970470; PMCID: PMC9371980; Jafari M, Pormohammad A, Sheikh Neshin SA, Ghorbani S, Bose D, Alimohammadi S, Basirjafari S, Mohammadi M, Rasmussen-Ivey C, Razizadeh MH, Nouri-Vaskeh M, Zarei M. Clinical characteristics and outcomes of pregnant women with COVID-19 and comparison with control patients: A systematic review and meta-analysis. Rev Med Virol. 2021 Sep;31(5):1-16. doi: 10.1002/rmv.2208. Epub 2021 Jan 2. PMID: 33387448; PMCID: PMC7883245.).

 We added a sentence in the results on this point.

    • Provide additional context or explanations for the statistical tests used to compare categorical and continuous variables between groups (e.g., Chi-square test, Mann-Whitney test).

                      We provided some clarifications in the methods in the revised manuscript (See l.178-181)

    • Clearly specify the significance levels used for determining statistical significance (e.g., p < 0.05) and any adjustments made for multiple comparisons.

 Significance level was already stated in the methods: All p-values below 0.05 were considered statistically significant. We did not correct for multiple testing because we did not want to test the hypothesis that all null hypotheses are true simultaneously as the Bonferroni correction would be. Rather, we explained why the tests of significance have been performed, and why (cf. Perneger TV BMJ 1998;316(7139):1236-6; doi: 10.1136/bmj.316.7139.1236).

discussion

    • Clearly interpret the study findings in relation to the existing literature. Discuss how the observed rates of maternal complications and psychiatric issues among women infected with SARS-CoV-2 during pregnancy compare to previous studies and what implications these findings may have for clinical practice.
    • Address any discrepancies or contradictions between the current study and previous research, such as the differences in findings compared to the German study mentioned.

We thank you for your comment . We added some comments in the discussion (See l.245-257)

    • Acknowledge the limitations of the study, such as the small sample size and low prevalence of outcomes, which may have limited the statistical power to detect significant differences between groups.

We thank you for your comment. We had already highlighted the limitations in the original manuscript (See l.258-278)

    • Provide a comprehensive comparison with existing literature on the impact of COVID-19 infection on maternal psychiatric issues. Discuss how the findings of the current study align with or diverge from previous research in this area.

We thank you for your comment . We added some comments in the discussion (See l.245-257)

    • Highlight the importance of investigating psychiatric outcomes across different phases of the pandemic to understand how the evolving COVID-19 situation may influence maternal mental health.

Thank you for your comment. We added a sentence in the revised manuscript See l.277-282

“Finally, maternal psychiatric outcomes should have varied across different phases of the pandemic and in function the vaccination against SARS-CoV-2. The results of our study should have been different if we have assessed women during the first wave of the COVID-19 pandemic and maybe the maternal psychiatric outcome would have shown higher rates of CB-PTSD and DPP because of higher perceived risks for maternal and neonatal complications in the beginning of COVID-19 pandemic.”

    • Emphasize the need for caution in interpreting the results and the importance of further research to confirm and expand upon the preliminary findings presented in the current study.

We thank you for your comment. We added a sentence in the revised manuscript (See l.287-88)

Reviewer 5 Report

Comments and Suggestions for Authors

I congratulate the authors on their choice of topic. I read the manuscript with great interest.

I would suggest considering the following recommendations:

Line 73 – please specify the time of COVID-19 pandemy - in 2021, the most dangerous strains of the Covid-19 epidemic had been already extinct

 Limitations of the study:

-          did you check the PCR-Covid test?

-          Did you test the group of women who were "negative"?
